# Novel Methodology to Recover Road Surface Height Maps from Illuminated Scene through Convolutional Neural Networks

**DOI:** 10.3390/s22176603

**Published:** 2022-09-01

**Authors:** Gonzalo de León, Julien Cesbron, Philippe Klein, Pietro Leandri, Massimo Losa

**Affiliations:** 1Department of Civil and Industrial Engineering (DICI), Engineering School, University of Pisa, Largo Lucio Lazzarino 1, 56126 Pisa, Italy; 2Joint Research Unit in Environmental Acoustics (UMRAE), Department of Planning, Mobility and Environment (AME), Université Gustave Eiffel, CEREMA, F-44344 Bouguenais, France; 3Joint Research Unit in Environmental Acoustics (UMRAE), Department of Planning, Mobility and Environment (AME), Université Gustave Eiffel-Lyon, CEREMA, F-69675 Lyon, France

**Keywords:** deep learning, convolutional neural networks, profilometer, image, photometric stereo, light, road pavements, texture, 3D reconstruction

## Abstract

Road surface properties have a major impact on pavement’s life service conditions. Nowadays, contactless techniques are widely used to monitor road surfaces due to their portability and high precision. Among the different possibilities, laser profilometers are widely used, even though they have two major drawbacks: spatial information is missed and the cost of the equipment is considerable. The scope of this work is to show the methodology used to develop a fast and low-cost system using images taken with a commercial camera to recover the height information of the road surface using Convolutional Neural Networks. Hence, the dataset was created ad hoc. Based on photometric theory, a closed black-box with four light sources positioned around the surface sample was built. The surface was provided with markers in order to link the ground truth measurements carried out with a laser profilometer and their corresponding intensity values. The proposed network was trained, validated and tested on the created dataset. Three loss functions where studied. The results showed the Binary Cross Entropy loss to be the most performing and the best overall on the reconstruction task. The methodology described in this study shows the feasibility of a low-cost system using commercial cameras based on Artificial Intelligence.

## 1. Introduction

Road surface characteristics have a major role in pavement’s life service conditions. Since the surface coarse is directly in contact with the vehicle tyres and exposed to the inclemency’s of the time, it is usually the first layer to exhibit issues. As a result, safety conditions [1], guidance comfort [2], tyre performance and noise emission [3] are among the principal affected surface characteristics.

The assessment of road surfaces is a matter of concern to both the public administration and the private sector, either for research interests or for monitoring purposes. In order to carry out the task, several techniques have been studied over the years. Nowadays, contactless techniques are preferred over classical methods due to their portability and high precision [4]. The high cost associated with the instrumentation makes them unaffordable for a broader public. Therein lies the importance to develop a simple, portable low-cost system that works with acceptable precision as an alternative solution to be adopted by these less favoured sectors. In this way, even if the economic resources are limited, the assessment of the road surface can still be made. Although they are both expensive, laser profilometers are slightly more affordable and therefore preferred over 3D lasers to measure surface texture, even though some important spatial information is missed [5].

In the recent years, the use of commercial photographic cameras for scientific purposes has been increasing [6], mainly due to the technological developments achieved in the sensor’s field [7] and their lower price compared with professional equipment. Among the different possibilities within the fields using imagery systems to retrieve depth information, three techniques satisfy the specific requirements for this present study: Photogrammetry, Depth from Focus/Defocus and Photometric Stereo.

In order to use Photogrammetry to retrieve depth information, matched points from different views are needed [8]. This procedure comes with two downsides. First, the matching algorithms and the iterative bundle adjustment required to solve the problem consumes too much computational time and resources. Second, in order to take the shots from different views two possibilities arises: to use several cameras or to develop a system that moves the camera around the study area. With either one or the other, the solutions would make the system too expensive and oversized, and therefore unpractical for our purpose.

Depth from Focus/Defocus is a technique where the images are obtained by actively controlling camera parameters, typically the focal setting or the image plane axial position, and taken from the same point of view. Afterwards, the depth estimation is retrieved by analysing the areas that are perfectly in focus at each focal distance [9]. The main drawback with this technique comes from the estimation of the focus metric. Particularly in the present paper, this technique is not pursued due to the difficulty to dynamically change the focal parameters with commercial cameras.

Photometric Stereo (PS) is a technique to retrieve depth information based on the reflectance properties of the material and their interaction with an incoming light source [10]. The principle of this method is to change the direction of the incident light source between successive shots while holding the viewing direction constant. This provides enough information to determine the surface orientation at each picture element. Since the geometry of the object does not change from the different shots, the correspondence between them is known a priori. Finally, the surface depth information is retrieved from the estimated normal map. In real set-ups, PS techniques carry on with different sources of errors such as: image noise, shadows occlusion, specular highlights, light calibration errors, inter-reflections or subsurface scattering of light which bias the normal and albedo estimation [11,12].

In the last decades, several studies supported the use of this technique for road surfaces problems [13,14]; nevertheless, they typically assess the performance of the reconstruction using overall surface descriptors, and the estimated surface is not compared with the true one.

The past 50 years have seen an exponential increase in the usage of computational resources for research purposes. Particularly, theoretical techniques such as Machine Learning (ML) and Deep Learning (DL) have seen the light after years of lacking resources to be feasible [15]. The recent evolution in the field of Graphics Processing Units’s (GPU) applied to modern artificial intelligence have placed the field within scientist practitioner’s reach.

Both ML and DL seek to solve an imposed task in an automatic manner [16]. The main difference lies in the fact that ML requires specific knowledge in the field to identify the features needed by the algorithm, while DL uses back propagation and gradient descent to make the network learn by itself which features are relevant and which ones are not. This specific characteristic have made the latter gain ground over the first one. Particularly, in the field of computer vision, one type of network has succeed to have impressive results, Convolutional Neural Networks (CNN) [17]. This type of network emerged from the study of the brain’s visual cortex, and they have been used in image recognition since the 1980s [18].

As stated before, several fields have started to use DL for research purposes and PS is no exception. In [19], DL was used to automatically learn the critical illumination conditions required at input. The network selects the most relevant illuminant directions used to estimate the surface normal. In [20], a novel photometric stereo network that directly learns relationships between the photometric stereo input and surface normals of a scene was introduced.

According to the Universal Approximation Theorem, any function can be approximated by a network with sufficient depth and enough amount of data. Generally, the scope of any Neural Network is to generalise the imposed task to best suit all the possible case scenarios. This creates a well-known problem known as “Over-fitting” [21], in which the network performs perfectly on the given dataset but poorly when new data is presented. In order to avoid this problem, the typical solution comes down to gather more data or to simplify the model.

The present work shows an end-to-end framework to extrapolate measurements carried out with a laser profilometer to a 3D surface using a commercial digital camera combined with different light sources. The task we achieve to solve is constrained to the single sample i.e., one model is developed for each sample. A CNN was trained, validated and tested on the created dataset. The results showed good scores both with global texture indicators and with the Pearson coefficient for spatial correlation. Besides its scientific novelty, this paper brings an alternative solution to assess road surfaces, with two major contributions. First, no expertise is needed in the field of photometric, computer vision, or numerical optimisation to obtain the surface reconstruction. Second, the economical benefit obtained by using a commercial camera instead of a professional instrumentation. This methodology could establish a new approach for texture measurement systems allowing a more affordable system for the least favoured sectors.

The remainder of this paper is structured as follows. Section 2 describes all the materials used in this work. Section 3 elucidates the methodology used to build the dataset with the corresponding pre- and post-processing. Afterwards, Section 4 presents the obtained results. Section 5 presents the discussions and implications of the previous sections. Finally, the conclusions are drawn in Section 6.

## 2. Materials

In the present work, an ad hoc dataset was created in order to be used to train a neural network. In the following sub-sections, the used materials are presented.

### 2.1. Samples

Two core samples extracted from in situ roads were used to train one model at a time. The surfaces studied are consistent with the most diffused typologies of surfaces in Italy. Table 1 shows the information regarding the samples and Figure 1 shows the two specimens.

Each sample had a diameter of approximately 145 mm due to the width of the core extractor and covered an area of 165 cm^2^. The samples were cleaned with water and a thin brush and left to dry naturally for two days. Four markers containing a checker pattern were added to each sample in order to use them as control points to fit a projective transformation matrix. Each pattern contains 10 × 7 alternated white and black squares, where each square side measures 1 mm; this yields a total of 54 control points per marker.

### 2.2. Camera and Lens

The camera used in this work is the Panasonic GX80-DCM which is among the most affordable camera’s in the market that satisfied the minimum requirements for geometrical and spatial resolution. The camera have a Four Thirds CMOS sensor (13 × 17.3 mm) with a resolution of 16 Megapixels. The sensor has the typical Bayer Pattern filter with the arrangement as follows: BGGR; the default Zoom Lens H-FS12032E 12–32 mm was used fixing the focal length to its minimum i.e., f-12 mm; ISO was set to 200, aperture fixed to an f-number equals to 3.5 and shutter speed to ¼. All the images were taken in raw format.

### 2.3. Laser

In this study a stationary profilometer laser was used to scan the profiles. The laser is manufactured by Greenwood Engineering and graded as Class 1 according with the ASTM E950 [22]. It is composed of a laser box containing the source and the receiver, a support frame allowing the box to move in a horizontal direction parallel to the sample surface, a displacement sensor, a control unit and management software. Table 2 summarises the characteristics of the equipment.

### 2.4. Lights

Four LED panels arranged in an 8 × 8 matrix were used with the scope to illuminate the sample from different directions. Each matrix contains a total of 64 White LEDs, where each light is individually addressable through a micro-controller board. Lights are 5050-sized LEDs with an embedded micro-controller inside that allows to set the brightness of each channel individually. Each LED can draw as much as 60 mA, yielding a total of 3.8 A per panel for full bright white. To supply the current to each matrix an external 5V DC power supply with an output current of 20 A was added. Additionally, a resistor between the micro-controller and the data input was placed in order to prevent spikes on the data line. The output is driven by a technique called Pulse Width Modulation (PWM) that works with a frequency of 1.2 kHz. This frequency sets the threshold for the shutter speed to 1/1000 before fluctuations in the light’s intensities captured by the sensor are noticed. Table 3 summarises the characteristics of the LEDs.

### 2.5. Convolutional Neural Network

A CNN was built and trained on a CPU with a processor Intel(R) Xeon(R) Gold 5118 @ 2.30 GHz with a NVIDIA Quadro P5000 GPU. The entire pre-processing of the images was developed in MATLAB. Once the dataset was ready, the training, validation and test of the network was done in Python using TensorFlow v2.9.

## 3. Methods

The scope of the present work is to develop a reproducible end-to-end methodology to obtain feasible results on the road surface estimation. The presented methodology comprises several multi-disciplinary fields ranging from signal and image processing to deep learning. It combines experimental data collection and pre-processing from two different domains i.e., images and laser profiles, and eventually combines them both using CNN. Figure 2 summarises the methodology.

As we can appreciate, the method consists of three main parts: The first one deals with the measurements of the samples using the profilometer laser and the linkage between the real-world and the image-world coordinates. The second part of the methodology deals with the development of a black-box with the scope to isolate the source of lighting. As it will be shown, with the same approach used in the first part, the corresponding coordinates inside the box are retrieved as well as the pixel intensities and the unitary vectors pointing towards the different light sources. Finally, the third step deals with the development of the architecture of the CNN and the optimisation of the hyper-parameters. The methodology developed in this section is the most important aspect of the study.

### 3.1. Camera Calibration

The camera calibration is an independent task that should be addressed before starting to use the instrument. This procedure allows to retrieve the intrinsic parameters of the camera, along with a number of coefficients that corrects for lens and sensor distortions using high order polynomial functions based on Brown’s model [23]. In this work, a professional software commonly used in photogrammetry was employed. The proposed technique requires the camera to observe a LCD screen projecting a checker-board pattern from different distances and perspectives. In order to obtain a robust calibration, ten sessions were carried out in different days, with the same camera set-up. Afterwards, for each correction coefficient and intrinsic camera parameter, a weighted mean ${\bar{x}}_{w}$ was calculated using the inverse of the squared errors as weights.
(1)$${\bar{x}}_{w}=\frac{\sum_{i=1}^{N}x_{i}/\varepsilon_{i}^{2}}{\sum_{i=1}^{N}1/\varepsilon_{i}^{2}}$$
where $x_{i}$ is the *i*-th parameter, $\varepsilon_{i}$ is the *i*-th associated error estimation and *N* is the total number of measurements and is equal to ten.

### 3.2. Laser

#### 3.2.1. Data Gathering

The laser was mounted on the transversal beam of the support frame and the camera was positioned zenithally from the sample using a fixed tripod as shown in Figure 3.

One at the time, the samples were placed in between the path of the laser line inside a special holder (element **5** in Figure 3) that allowed to rotate the samples around their z-axis while the x-y plane was fixed. Before starting the procedure, the initial and ending point of the session must be defined. In order to do so, two discontinuities close to the surface within the profile path were introduced in a way that the distance between the beginning and end of the profile matched 120 mm. Regarding the respective initial and ending points in the image coordinate system, the laser was manually placed at the mentioned position and a shot was taken for each one of them. This procedure was repeated twenty times in order to minimise errors. Afterwards, in post-processing, an optimisation procedure was carried out to estimate the best initial and ending positions by minimising the distance between the real-world measurement and the estimated distance obtained using the transformation matrix. Once our reference system had been defined, the measurement process takes place.

At each position three profiles were acquired using the profilometer laser and one image was captured. Subsequently, the sample was rotated approximately one degree and the process repeated until the end. In this way, the relative position for each rotation was retrieved through the transformation matrix using the markers. At the end of the procedure, a total of 180 positions were registered and a total of 540 profiles for each sample were acquired.

#### 3.2.2. Post-Processing

Regarding the signal workflow, in this work a slight modification was applied to the procedure suggested by the ISO 13473-1:2019 [24]. Table 4 illustrates the workflow applied to the signal in the present work and their differences with the ISO.

The laser’s software provides an output already correcting the drop-out missing values and re-sampling the signal to 0.1 mm. Next, for the spike identification and outliers removal, two approaches were combined sequentially. The first one is a k-means based algorithm, while the second one is based on the determination of the zones with fast variation of the signal using the wavelet transform. The first outlier removal technique is a statistical-based approach that allows to eliminate errors coming from the manual displacement of the laser box, yielding a more robust estimate of the true profile. The procedure first estimates the mean of the three runs. Then, it calculates a point-wise distance from each signal to the mean. Finally, it discards the furthest point and calculates the mean of the two remaining points. The second outlier removal is a wavelet-based approach. The wavelet transform can be used to determine the fast variation of a signal. The wavelet transform of x(t) is written as: (2)$$X_{\psi }\left(u,s\right)=\int_{-\infty }^{\infty }x\left(t\right)\frac{1}{\sqrt{s}}\bar{\psi }\left(\frac{t-u}{s}\right)dt$$
where $\psi \left(t\right)$ is the wavelet and $\bar{\psi }\left(t\right)$ its complex conjugate, *u* and *s* are the shifting and scaling parameters respectively.

In this work, the Mexican Hat wavelet was chosen as suggested by [25]. The choice of the wavelet determines a threshold value which is a function of the wavelet itself and the RMS value of the upper part of the signal. All points for which the wavelet transform value is over the threshold as well as their two adjacent points are removed and replaced by a linear interpolation between the closest valid readings. Figure 4 shows the identified peaks in the upper part of the signal while Figure 5 displays the corresponding wavelet transformation, the threshold value and the identified outliers.

Afterwards, since the signal was considerably short, the removal of long-wavelength components was ignored and only the slope suppression was performed. Finally, the signal was normalised between zero and one in order to be used as ground truth data by the CNN.

#### 3.2.3. Reference System

As mentioned above, for each laser measurement, one image was captured. This procedure was conceived for two reasons. First, the relative position of each laser measurement with respect to the reference system must be obtained; this can be accomplished using the fixed markers placed at the samples. Second, the transformation matrix between any two positions can be calculated, and therefore any measurement at any position can be projected to another. This leads to an important practical conclusion: all the positions can be projected into a single one.

In matrix representation, we can represent a projectivity as follows: (3)$$x_{proj}=Px$$
where *P* is the *Transformation Matrix* and is represented by a matrix chain as follows: (4)$$P=\left[\begin{array}{cc} sR & t\\ 0^{⊤} & 1 \end{array}\right]\left[\begin{array}{cc} k & 0\\ 0^{⊤} & 1 \end{array}\right]\left[\begin{array}{cc} \lambda & 0\\ 0^{⊤} & 1 \end{array}\right]\left[\begin{array}{cc} I & t\\ \nu^{⊤} & \nu \end{array}\right]$$
where the first term is a 4 DOF similarity matrix, the second term is a 1 DOF shear transformation matrix, the third term is a 1 DOF scaling term and finally the last term is a 2 DOF elation matrix.

Figure 6 shows the sample at positions 0 and 45 with the mentioned markers. The measured profile at the given position is displayed with an horizontal line, i.e., orange line in the left image and yellow line in the right. After the transformation is fitted, the 45th line can be projected into the 0th image and vice versa.

Figure 7 present a plot of the same line extracted from two different images at two different positions. The x-axis shows the distance in pixels while the y-axis stands for the normalised intensities. At the top right corner, the figure displays the cross-correlation for the two extracted signals, showing that the maximum correlation between the signals is when the lag is zero. The small differences noticed between the signals are due to the change of the light conditions between sample positions.

Figure 8, Figure 9 and Figure 10 exhibit a triplet graph, where the first two images show position 0 and 45, respectively, while the third one displays an overlapped image of them using a composite image with three different colour bands. Green regions highlight higher differences in position 0 than 45, while magenta does the opposite. Grey regions in the composite image show where the two images have the same intensities. It is evident from Figure 9 and Figure 10 that the fitted transformation synchronises the images almost perfectly.

### 3.3. Lighting Box

In this study a special lighting box was constructed with the scope to isolate the samples from environmental lighting and therefore to have only the contribution from direct light sources. The building process followed the Fast Development Cycle suggested by [26]. The structure of the box was made entirely of wood. The dimensions of the box were carefully designed in order to achieve the desired optical scale for the camera. In the top face, a hole with the dimension of the lens diameter was drilled with the purpose to host it. Even though the hole was drilled accurately, a millimetric space between them still exists; therefore, to fully block the incoming light a high-density polyurethane foam was placed around the lens and the hole. All the interior walls, ceiling and floor of the box were covered with light absorbing fabric in order to avoid unwanted reflections from the walls.

Regarding the lighting sources, four Arduino™ LEDs panels were used. Particularly, only the central four LEDs were activated yielding a smaller 4 × 4 matrix. The position of the LED was chosen to be perpendicular to a 175 mm line starting from the centre of the sample with an elevation angle of 60° and azimuth angles equally distributed separated by 90°. Figure 11 shows an open view of the box design without the interior fabric cover.

#### 3.3.1. Data Gathering

The camera was positioned in the top hole and fixed to the Polylactic Acid (PLA) piece holder. Next, the samples were introduced inside the box from the bottom hole and positioned at the working distance with the help of a lifting platform.

The LED panels were programmed to turn on sequentially. For each panel only the central 16 LEDs were displayed simultaneously and one shot per light position was taken. Additionally, an extra photo was taken with all the four lights simultaneously turned on in order to use it later to obtain the transformation matrix.

Finally, the entire process was repeated 360 times rotating the sample approximately half degree at the time. In this way, at the end of the procedure, 1440 photos were collected, each one of them containing information regarding one position and one incoming light direction. Remembering that in the first phase we collected the positions for 180 lasers’ profiles, now we have for each laser profile 360 different incoming lights. This yields us a total of 64,800 instances per pavement sample.

#### 3.3.2. Post-Processing

Finally, following the logic presented in Section 3.2.3, we can obtain the coordinates of the measured laser profiles inside the box. To avoid redundancy we only show Figure 12 displaying the triplet after the transformation has been applied. Since the background is now black, the differences in the surrounding part of the sample are all green.

### 3.4. Light Calibration

The light directions of the sources were calibrated following the procedure suggested in [27], where the vectors pointing to the source are retrieved using a planar mirror containing a pattern of known dimensions. The technique assumes that the image is formed by perspective projection and the intrinsic parameters of the camera are known. Using the procedure described in [28], we retrieve the two-plane homography which relates the scene points to the image points. In this way, we obtain for each specular point at the mirror a vector pointing to the source. Repeating this procedure for several orientations of the mirror, the point intersecting the lines is retrieved by solving an over determined system using the Moore–Penrose Pseudo-inverse.

In this study, each individual LED direction was calculated separately, meaning that 16 light directions were obtained per panel and a total of 64 light directions calculated. The procedure was carried out using ten different positions of the planar mirror per session. Finally, for a robust estimation of the parameters, five sessions were carried out and the weighted mean calculated.

### 3.5. CNN

Considering a dataset composed by two random variables $\left(X,Y\right)\in R^{n}$, where *X* denotes the pixel intensity distribution in the area of interest and *Y* the correspondent height value, then if we hypothesise that *Y* is a deterministic function of *X*, the problem reduces to find the function *f* mapping f:X→Y that minimises a cost function L.

As mentioned before, an important difference between this work and usual approaches in DL is that the task imposed to the network is not to generalise to other unseen samples, but rather to extrapolate for each sample the obtained measurements with the profilometer laser to the captured scene illuminated from different light positions. Therefore, one model was built for each sample.

Once the dataset was shaped, cleaned and correctly normalised the data was subdivided into the corresponding train (90%), validation(5%) and test (5%) sets. There is no general rule for the data splitting ratio. The splitting percentage has an influence on the variance. With less testing data, the statistical performance of the model has a greater variance and if less training data is used the variance on the estimated parameters increases.

The choice of the splitting ratio was validated by checking the distribution similarity of the three sets. Therefore, the three sets can be assumed equally representative of the entire population, and the biggest amount of data is being used to train the model.

Each set contains the corresponding tensors pairs of input (features) and ground truth (measured). The train set is used by the algorithm to learn by minimising the defined cost function, while the validation set is used to determine the optimal architecture and tune the hyper-parameters of the network. On the other hand, the test set is separated and never seen by the model. At evaluation time, we compare the estimated reconstruction with the true measured profile obtained with the profilometer.

A typical technique to avoid over-fitting is data augmentation. This allows to virtually increase the size of the dataset and it is only used during inferring time i.e., only to train and validation sets. In this paper, *Random Flipping* was used.

The *Learning Rate* was set to 0.01 to start and then adjusted during training phase using a *Learning Scheduler* that reduces the *Learning Rate* between epochs when the validation metric stop improving. The *Learning Rate* was updated by a factor of 0.1 after 5 epochs without improvement. The *Batch Size* was chosen to be 8, which is a relatively small value compared with typical values used in DL. The chosen *Optimiser* was the Nesterov-accelerated Adaptive Moment Estimation (*Nadam*) algorithm which is an extension to the Adaptive Movement Estimation (*Adam*) optimisation algorithm to add Nesterov’s Accelerated Gradient (*NAG*) or Nesterov momentum.

The network hyperparameters are summarised in Table 5.

The architecture of the proposed network is given by Figure 13.

The network has 10 layers. First, in parallel, one branch performs a depth-wise convolution, i.e., a single convolutional filter per each input channel, while the other branch multiplies the input intensities *I* in a point-wise manner by the inverse matrix of the light source vector *S*. The matrix *S* contains the unitary vectors pointing to each corresponding light source. All the vectors lay within a plane resting at the top face of the surface. Next, three sequential layers concatenate the results, reduce the depth with a point-wise convolution and reshape the output. Afterwards, 10 filters per branch with variables dimensions are convoluted in parallel, the kernel sizes are shown as a footnote in Table 5. These dimensions of the kernels were chosen to resemble the dimension of the bandwidths used for spectral analysis in 1/1 octave-band. Then, three sequential layers of concatenation, point-wise and reshape were appended. Finally, two fully connected dense layers were placed, where the first one have a drop out rate of 20%. All the selected activation functions are displayed in Table 5.

Three different loss functions were studied, the Mean Squared Error (MSE), the Mean Absolute Error (MAE) and the Binary Cross-Entropy (BCE) [29]. Since their use is quite diffuse in the field of DL, the formulation for each loss function is omitted. The MSE penalises large prediction errors and therefore is more sensitive to outliers. On the other hand, the MAE is more robust to outliers and yields results closer to the median, which due to the nature of road profiles is a desired result. BCE is a measure of dissimilarity between two distributions and is composed by two terms. While the first rewards correct estimations, the second penalises the wrong predictions for each class. Using the BCE to frame the reconstruction task as a multi label binary classification problem, where each pixel intensity represents the probability that the pixel should be black, makes the model converge faster [30]. The model was trained for a total of 100 *Epochs* i.e., a complete loop of the entire dataset. Finally, in order to assess the performance of the model, six global texture indicators where used as metrics to analyse the results, MSD, Arithmetic Mean Height ($R_{a}$), Root Mean Square Height ($R_{q}$), Skewness ($R_{sk}$), Kurtosis ($R_{ku}$) and Root Mean Square Gradient ($R_{dq}$). The metrics are widely used in the field and their definition can be found in [24] for the MPD and [31] for the others. For the MSD, the considered segment length was 12 [cm], slightly over the 10 [cm] suggested by the ISO.

## 4. Results

The first step is to analyse the evolution of the different loss functions during inference. This is displayed in Figure 14. Since the scale between the losses is different, the normalised loss is shown. The x-axis represents the evolution in *Epochs*, while the y-axis shows the normalised loss. The continuous lines represent the training set for each loss, while the dashed lines stand for the validation set. The three losses converges eventually to zero values on the train set, but since the normalised loss is considered, absolute minimum values are not appropriate to be used for analysis. Therefore, the indicators of performance during inference are the speed of convergence and the distance between the training and validation set. It is clear from the graph, that the *BCE* is the most performing loss during the training phase, being the fastest to converge and showing the smallest distance between validation and training.

Regarding the test set, as mentioned before, the losses have different scales and direct comparison is not possible; therefore, global texture indicators were used to analyse the results. Table 6, Table 7 and Table 8 show the mean of the difference between the estimated metric and the true value for the *MSE*, *MAE* and *BCE*, respectively. Again, it can be seen that the *BCE* yields lower mean differences for all the metrics in consideration, indicating a better performance against the others.

However, is quite easy to obtain close values for two different profiles of the same sample. This can be appreciated in Figure 15, where the image shows two different profiles from the same sample, and in Table 9, where the correspondent indicators are calculated.

Consequently, the Pearson coefficient was chosen to assess the spatial correlation existing between predicted and measured profiles. The mean and the standard deviation on the test set of the Pearson’s correlation coefficient are shown in Table 10. Again, the *BCE* shows the highest correlation mean and the lowest standard deviation with values equals to μ=0.9198 and σ=0.0422.

Figure 16 and Figure 17 display a comparison of the reconstructed profiles using the *BCE* loss against the ground truth measured profile for samples 1 and 2, respectively.

Finally, Figure 18 shows three zoom-in portions of the smoothed three-dimensional reconstruction of the dense graded surface. The axis is only expressed for the bigger portion in the left part of the image. The x- and y-axis are in 10th of mm while the z-axis is represented in mm.

## 5. Discussion

The discussion of the pitfalls for the chosen materials and the created dataset are presented in Section 5.1. Then, some remarks on the methodology are drawn in Section 5.2. Finally, extra analysis of the obtained results are presented in Section 5.3.

### 5.1. Materials

The use of in situ extracted samples is not compulsory for the method but provides a convenient and safe workspace. In our particular case, since the method is expected to be further developed to be used for monitoring purposes, the use of representative samples as close as possible to the expected future samples was important. The use of custom made samples in the laboratory would be also possible for investigation purposes, but it would increase the complexity of the model since the Lambertian assumption could fail due to the specular properties of virgin binder. On the other hand, this could simplify the gathering procedure and increase exponentially the available amount of data. Moreover, the similar reflective properties the surfaces would have, could help to generalise the model to different surface types.

The economical cost associated with the profilometer laser used to sample the ground truth data is probably the most limiting aspect for the dataset creation, but compared against the cost associated with a 3D laser scan, the scale tips in favour of the laser. On the other hand, if a 3D laser scan would have been available, the amount of data and the spatial continuity of it, would have been better. Also, the use of markers and the entire procedure to register the matching points to retrieve coordinates could have been avoided.

The photographic camera was chosen based to its associated practical characteristics, i.e., compactness, simplicity to use, easy GUI, connection and remote control. Nevertheless, zoom-lenses cameras are often discouraged for photometric purposes due to their calibration instability. Therefore, the dataset creation with the camera should be performed within a short window of time after the calibration is carried out. For the calibration, several sessions are advised and disabling all “smart” features that commercial cameras often offer is compulsory. It is important to work with RAW format instead of the converted formats that the commercial camera offers. The reason is that non-linear corrections are applied to the intensity values to make them look “nicer” to human perception.

Regarding the LED lights, a commercial brand was chosen due to the simplicity offered for connections and the coding flexibility to set-up the intensity magnitude. The latter is extremely useful for an easy configuration and maximisation of data. All the requirements and connection settings are supplied by the manufacturer as well as the intrinsic characteristics of the lights. The authors do not encourage the use of specific brands nor discourage the development of a custom system for the lighting.

Finally, for the development of the Convolutional Neural Network, training and full deployment of the model require a GPU. Fortunately, nowadays, there are some providers with limited access free of charge that allows to use computing resources including GPUs and TPUs.

### 5.2. Methodology

The presented methodology proposes an alternative solution to recover road surface elevation maps from images taken with a commercial camera and different light sources. The scope of this study was to prove the feasibility of the method.

The advantages of the proposed methodology is mainly linked with the minimum knowledge required in the field of photometric, computer vision or numerical optimisation to solve the surface reconstruction problem. On the other hand, the main drawback with the current proposed methodology is associated with the practicality of it, i.e., one model is required to be trained per sample. In addition, the required equipment to measure the ground truth surface and the amount of time needed to gather the data can be questioned since they are only needed to train the model but, as already stated, a new network have to be trained for each sample. This opens the door to future investigation and the necessity to develop a more generalised model.

### 5.3. Results

The results were previously introduced showing how the *BCE* had the best performance. The small distance observed in Figure 14 between validation and training is an indicator of an unbiased model. Meaning that the results are expected to generalise quite well on unseen data. Nevertheless the test set is set aside for this purpose.

The results in Table 6, Table 7 and Table 8 exposed how these indicators are not appropriate for reconstruction purposes and they are more suitable for more generic assessment of the sate of a surface or for classification purposes. Nevertheless, these indicators are widely accepted in the field and several studies used them. In addition, this parameters are insensitive to the spatial correlation, meaning that if a shifted or flipped version of a profile is analysed with these global metrics the same value would be obtained for both of them. This exposes the necessity to use a different metric since global indicators by itself are poor descriptors for similarity reconstruction analysis for road surfaces.

It was not possible to compare the obtained results with other studies working with photometric deep learning for road surfaces, due to the unavailability of parameters weights to build and run the models. Another constraint to compare against different methods lies in the fact that most of the current DL models require different input information or the expected ground truth are the normal directions. In the latter case, the ground truth used in this methodology corresponds to laser profiles and the orientation of the faces are not provided; therefore, the normal method only has one component compared to the three expected for the other methods.

Therefore, the reconstructed profiles were compared against the measured ones. Particularly, sample ID 1 is a dense-graded road surface with grain sizes ranging from 0 to 6 mm, which makes the surface quite closed and therefore a lower amplitude on the profile is perceived. On the other hand, sample ID 2 is a gap-graded surface with grain sizes ranging from 0 to 12 mm, which provides the surface a bigger height amplitude compared with sample ID 1. This different nature on the profiles makes the model for the dense pavement to perform slightly better. Regarding the gap-graded sample, the depression zones of the profile are the regions where the model performs worst. This could be attributed to the light occlusion due to the shape of the profile. In order to overcome this problem, lights with different wavelengths are expected to be studied, hoping that lights with larger wavelengths could penetrate further in to the gap-graded samples.

Finally, the 3D reconstruction was shown. Indeed, it should not be considered as 3D surface since each row and column from the image were considered as single profiles, and a more precise term would be elevation raster. Small discrepancies occurred between some adjacent profiles and, therefore, a Gaussian filter was applied to the reconstruction to improve the appearance for the view. This effect could be attributed to the noise introduced by the CNN and opens the door for future investigations in order to improve the current state of the model.

## 6. Conclusions

In the present paper, we started from the premise to develop a fast and low-cost system as an alternative solution to assess road surfaces for the least favoured sectors. The purpose of the study was to develop a methodology to obtain a faithful reconstruction of the 3D road surface. In order to do so, extra information coming from a sensor was needed. Keeping in mind the fast and low-cost objective, a commercial photographic camera was chosen due to its associated practical characteristics, i.e., compactness, simplicity to use, easy GUI, connection and remote control.

Due to the nature of the new sensor, radiometric concepts were studied to have a better understanding on the laying phenomena of the interaction between the light and the surface. From this study, photometric stereo came out to be the best fit for our case-study. Therefore, a closed black-box was constructed with the scope to control the lighting conditions. However, complex numerical methods are required to solve the problem. Consequently, neural networks came out as a solution to avoid complex modelling and expensive computational time. The imposed task to the neural network was to extrapolate measurements obtained with a class 1 profilometer laser to the rest of the scene illuminated with light sources positioned around the surface sample and captured with the camera sensor. The chosen type of neural network was a CNN due to their advantages compared against other types i.e., fast and less parameters.

The data was needed to be structured in pairs (*input*, *ground truth*), which means that the correspondence between the image and the real world coordinates had to be found. For this reason, four patterns with known dimensions were printed and attached on a small flat surface over the sample. Afterwards, in post-processing, a transformation matrix was fitted allowing to obtain the correspondence between them. The corresponding images intensities and light directions per pixel were given as input and the measured laser profiles as ground truth.

At inference time, three different loss functions were studied, *MSE*, *MAE* and *BCE*. Results showed a faster convergence and better generalisation on the training and validation set for the *BCE*. In order to assess the results for the test set, a group of global surface metrics were analysed. The outcome showed that the results obtained with the proposed method are practically identical when these indicators are used, This entail an important practical application: the estimated indicators are practically the same as the ones calculated with the professional instrument, showing that the system yields excellent results for general assessment when global metrics are required. However, they were not sufficient to evaluate a proper reconstruction. Therefore, the Pearson correlation coefficient was used. The results proved that the *BCE* still outperformed the other losses and that faith-full reconstructions were obtained.

Finally, a raster elevation reconstruction was carried out showing the necessity to perform a Gaussian smoothing due to discontinuities obtained in some adjacent estimated profiles.

Aside from the scientific novelty for the proposed method using CNN’s, this paper brings an alternative solution to assess road surfaces, with three major contributions. First, no expertise is required in the field of photometrics, computer vision or numerical optimisation to obtain the surface reconstruction. Second, the associated economical reduction obtained using a commercial camera instead of a professional instrumentation. Third, the high accuracy achieved is remarkable. However, the vast amount of data needed for the creation of the dataset as well as the associated time to train the network can be a significant drawback. This encourages researchers to pursue a better generalised model that works for all type of surfaces simultaneously. Nevertheless, the proposed methodology would stand, setting the base for this future work. This methodology could establish a new approach for texture measurement, allowing a more affordable system for the least favoured sectors.

## Figures and Tables

**Figure 1 sensors-22-06603-f001:**
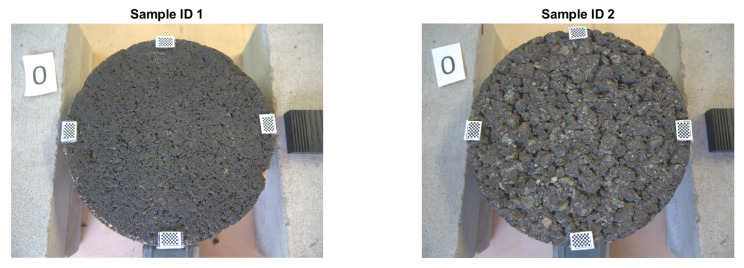
Samples. **Left:** Sample ID 1—Dense Graded. **Right:** Sample ID 2—Gap Graded.

**Figure 2 sensors-22-06603-f002:**
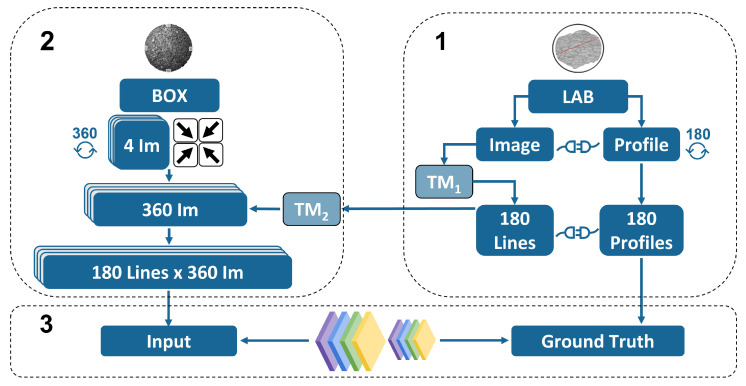
Overall methodology chart. (**1**) Laboratory laser profiles. (**2**) Images inside box. (**3**) CNN.

**Figure 3 sensors-22-06603-f003:**
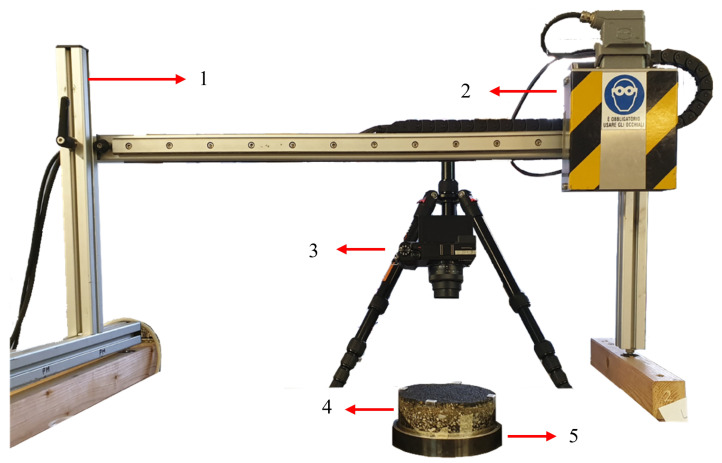
Experimental Set-Up. From Top to Bottom: (**1**) Fixed Beam. (**2**) Laser Box. (**3**) Camera and Tripod. (**4**) Sample. (**5**) Sample Holder.

**Figure 4 sensors-22-06603-f004:**
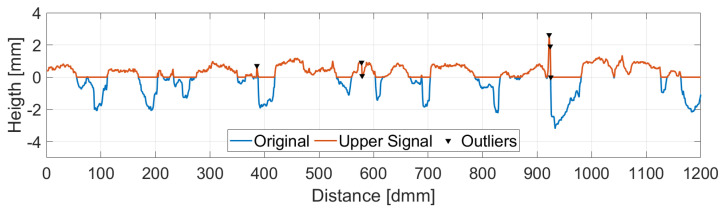
Outlier Detection. Upper Part vs. Original Signal.

**Figure 5 sensors-22-06603-f005:**
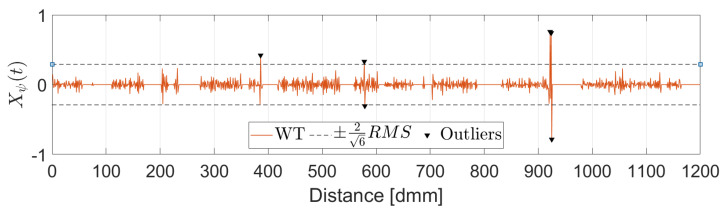
Outlier Detection. Wavelet Transformation.

**Figure 6 sensors-22-06603-f006:**
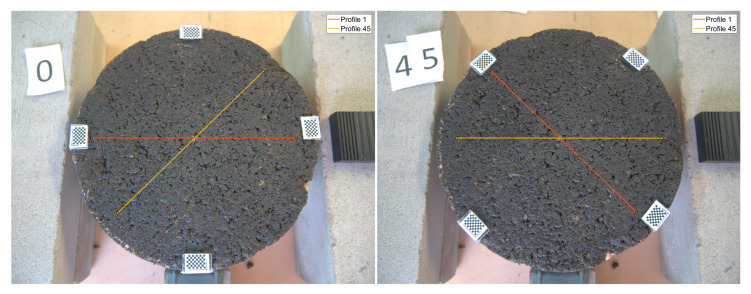
Projection of measurements at different positions. **Left:** 0th position, measured profile in orange (horizontal), projected profile in yellow (45°). **Right:** 45th position, measured profile in yellow (horizontal), projected profile in orange (135°).

**Figure 7 sensors-22-06603-f007:**
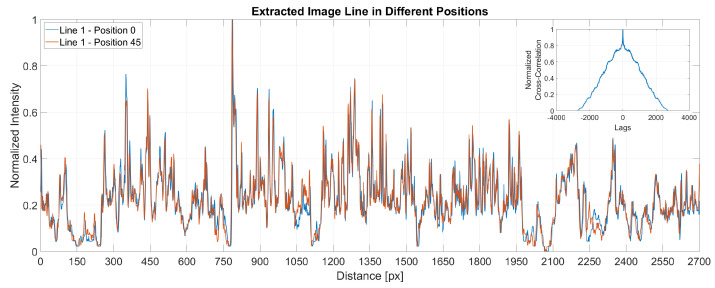
Cross–correlation between samples.

**Figure 8 sensors-22-06603-f008:**
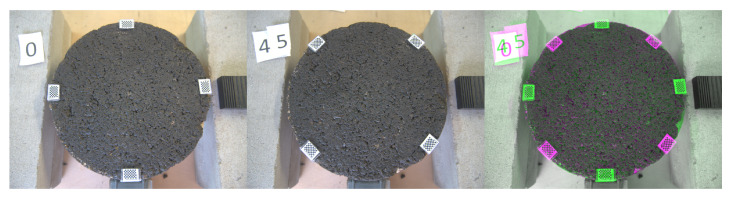
Triplet image. **Left:** 0th position. **Centre:** 45th position. **Right:** Overlapped before transformation.

**Figure 9 sensors-22-06603-f009:**
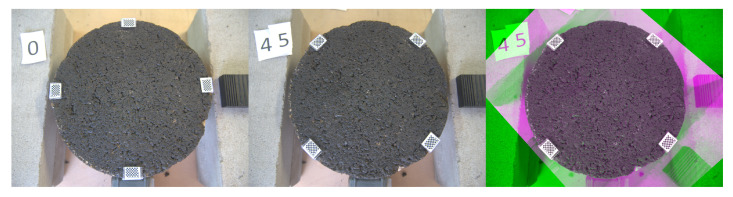
Triplet image. **Left:** 0th position. **Centre:** 45th position. **Right:** Overlapped after transformation 0th→45th.

**Figure 10 sensors-22-06603-f010:**
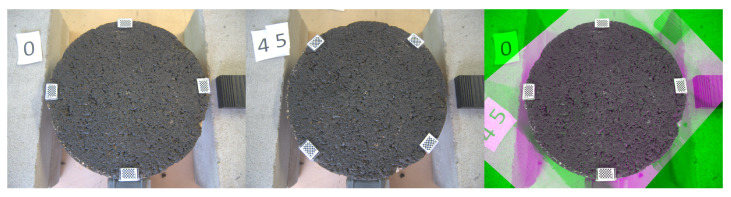
Triplet image. **Left:** 0th position. **Centre:** 45th position. **Right:** Overlapped after transformation 45th→0th.

**Figure 11 sensors-22-06603-f011:**
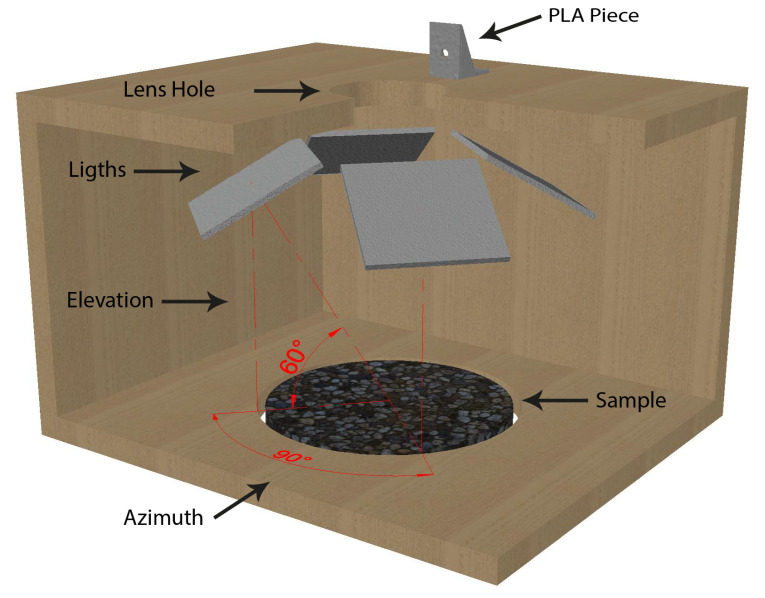
Interior box design.

**Figure 12 sensors-22-06603-f012:**
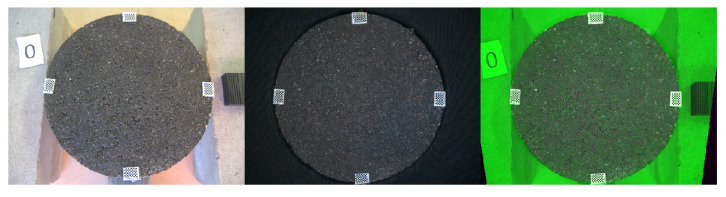
Triplet image. **Left:** 0th position in reference system. **Centre:** 0th position inside the box. **Right:** Overlapped after transformation Ref. Sys.→ Box.

**Figure 13 sensors-22-06603-f013:**
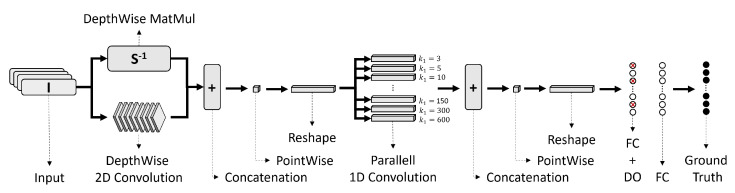
Proposed CNN Architecture.

**Figure 14 sensors-22-06603-f014:**
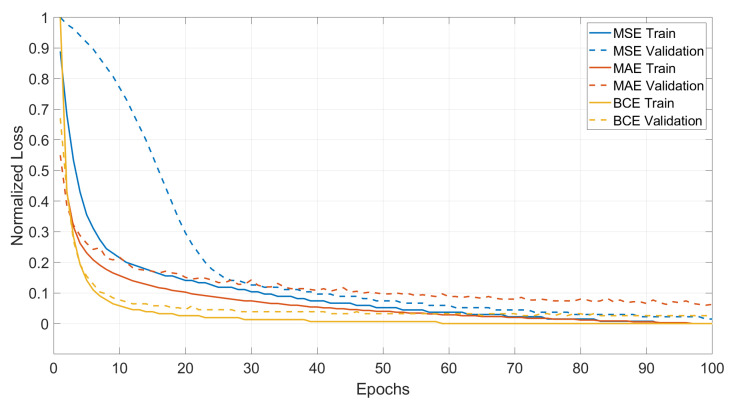
Training vs. Validation.

**Figure 15 sensors-22-06603-f015:**
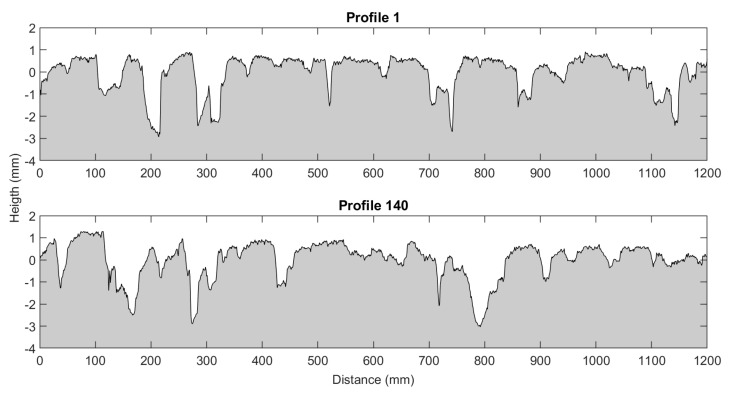
Two profiles from the same sample.

**Figure 16 sensors-22-06603-f016:**
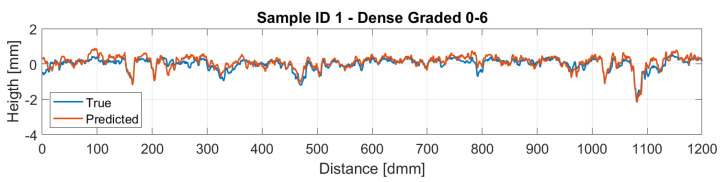
Sample 1 Reconstruction.

**Figure 17 sensors-22-06603-f017:**
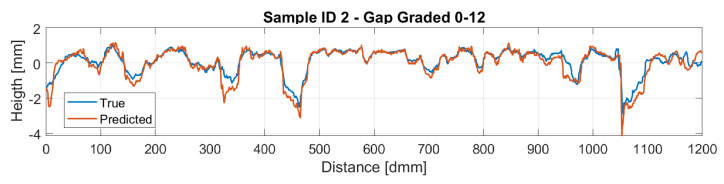
Sample 2 Reconstruction.

**Figure 18 sensors-22-06603-f018:**
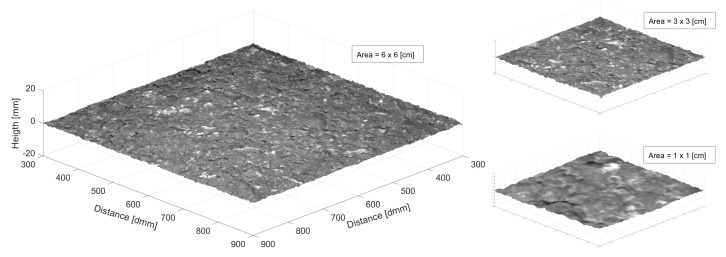
Three–dimensional reconstruction at different scales. **Left:** 6 × 6 cm. **Top Right:** 3 × 3 cm. **Bottom Right:** 1 × 1 cm.

**Table 1 sensors-22-06603-t001:** Samples Types.

Sample ID	Type	Grain Size [mm]
1	Dense Graded	0–6
2	Gap Graded	0–12

**Table 2 sensors-22-06603-t002:** Characteristics of Profilometer Laser.

Device	Type	Wavelength	Sampling Rate	Sampling Interval	Resolution	Measuring Speed
Stationary	Contact-less	B-D	16 kHz	0.1 mm	0.05 mm	Slow

**Table 3 sensors-22-06603-t003:** Characteristics of LEDs.

Colour	Wavelength Range	Temperature White	Single LED Dimension
Warm White	425–725 nm	2700–3000 K	5 × 5 mm

**Table 4 sensors-22-06603-t004:** Laser workflow.

Steps	Changes from ISO
Drop out correction and interpolation	Same as ISO
Re-sampling	Same as ISO
Spike identification and outlier removal	Different from ISO
Removal of long-wavelength components	Only slope suppression
Normalisation	Not in ISO

**Table 5 sensors-22-06603-t005:** CNN Architecture.

Learning Rate	Batch Size	Optimiser	Layers	Type	Activation Function	# Kernels	Kernel Size
0.01	8	Nadam	$1_{a}$	DWC2D ^1^ + BN ^2^	ReLu	10	3 × 1
			$1_{b}$	PWMM ^3^	-	-	-
			2	Concatenate	-	-	-
			3	PointWise	ReLu	1	1 × 1
			4	Reshape	-	-	-
			5	Parallel C1D ^4^	ReLu	90	*k* * × 1
			6	Concatenate	-	-	-
			7	PointWise	ReLu	1	1 × 1
			8	Reshape	-	-	-
			9	FC ^5^ + DO ^6^	ReLu	-	1200
			10	FC	Sigmoid	-	1200

^1^ DepthWise Convolutional 2D. ^2^ Batch Normalisation. ^3^ PointWise Math Multiplication. ^4^ Convolutional 1D. ^5^ Fully Connected. ^6^ Drop Out. * *k* = *3*, *6*, *10*, *20*, *40*, *75*, *150*, *300*
*600*.

**Table 6 sensors-22-06603-t006:** Mean difference on test set with MSE.

	MSD	$R_{a}$	$R_{q}$	$R_{\mathrm{sk}}$	$R_{\mathrm{ku}}$	$R_{\mathrm{dq}}$
**Train**	1.46 × 10${}^{-2}$	1.40 × 10${}^{-4}$	1.47 × 10${}^{-4}$	3.99 × 10${}^{-4}$	3.70 × 10${}^{-3}$	1.11 × 10${}^{-4}$
**Valid**	1.50 × 10${}^{-2}$	1.48 × 10${}^{-4}$	1.52 × 10${}^{-4}$	4.11 × 10${}^{-4}$	3.80 × 10${}^{-3}$	1.13 × 10${}^{-4}$
**Test**	1.50 × 10${}^{-2}$	1.57 × 10${}^{-4}$	1.64 × 10${}^{-4}$	4.16 × 10${}^{-4}$	3.85 × 10${}^{-3}$	1.15 × 10${}^{-4}$

**Table 7 sensors-22-06603-t007:** Mean difference on test set with MAE.

	MSD	$R_{a}$	$R_{q}$	$R_{\mathrm{sk}}$	$R_{\mathrm{ku}}$	$R_{\mathrm{dq}}$
**Train**	2.49 × 10${}^{-2}$	1.73 × 10${}^{-4}$	2.00 × 10${}^{-4}$	1.08 × 10${}^{-4}$	9.32 × 10${}^{-3}$	3.16 × 10${}^{-4}$
**Valid**	2.51 × 10${}^{-2}$	1.93 × 10${}^{-4}$	2.23 × 10${}^{-4}$	1.09 × 10${}^{-4}$	9.41 × 10${}^{-3}$	3.21 × 10${}^{-4}$
**Test**	2.49 × 10${}^{-2}$	1.91 × 10${}^{-4}$	2.24 × 10${}^{-4}$	1.10 × 10${}^{-4}$	9.57 × 10${}^{-3}$	3.24 × 10${}^{-4}$

**Table 8 sensors-22-06603-t008:** Mean difference on test set with BCE.

	MSD	$R_{a}$	$R_{q}$	$R_{\mathrm{sk}}$	$R_{\mathrm{ku}}$	$R_{\mathrm{dq}}$
**Train**	6.80 × 10${}^{-3}$	9.65 × 10${}^{-5}$	9.72 × 10${}^{-5}$	2.32 × 10${}^{-4}$	2.18 × 10${}^{-3}$	5.94 × 10${}^{-5}$
**Valid**	7.03 × 10${}^{-3}$	1.09 × 10${}^{-4}$	1.08 × 10${}^{-4}$	2.42 × 10${}^{-4}$	2.28 × 10${}^{-3}$	6.02 × 10${}^{-5}$
**Test**	7.22 × 10${}^{-3}$	1.07 × 10${}^{-4}$	1.06 × 10${}^{-4}$	2.53 × 10${}^{-4}$	2.36 × 10${}^{-3}$	6.28 × 10${}^{-5}$

**Table 9 sensors-22-06603-t009:** Geometrical texture indicators.

	MSD	R_a_	R_q_	R_sk_	R_ku_	R_dq_
**Profile 1**	0.9259	0.6209	0.8142	−1.6012	4.8470	0.1187
**Profile 140**	0.9302	0.6015	0.7997	−1.5215	4.8369	0.0923

**Table 10 sensors-22-06603-t010:** Mean Pearson Correlation on test set.

${\bar{r}}_{\mathrm{MSE}}$ μ±σ	${\bar{r}}_{\mathrm{MAE}}$ μ±σ	${\bar{r}}_{\mathrm{BCE}}$ μ±σ
0.8954 ± 0.0463	0.8407 ± 0.0885	0.9193 ± 0.0422

## Data Availability

The data presented in this study are available on request from the corresponding author.

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
