# Peer review of "Novel Methodology to Recover Road Surface Height Maps from Illuminated Scene through Convolutional Neural Networks"

_sensors, 2022, doi:10.3390/s22176603_

Round 1

Reviewer 1 Report

This paper introduce a novel method to recover road surface height maps through photometric stereo and CNN. Comprehensive experiments have been conducted for dataset creation and design. The CNN architecture was introduced, where three loss functions were tested on their performance. Overall, this paper is interesting and fit the scope of Sensor Journal. Nevertheless, I suggest to solve the following issues before it is to be accepted for publication.

1. I suggest to give a framework of the methodology.

2. I suggest to move 2.5. Convolutional Neural Network to the Method Section.

3. In Subsection 3.2.2 for post-processing. You introduced two filter methods, i.e., the K-means method and the Mexican Hat wavelet method, where the latter is selected according to previous publication. That means, the K-means method contributes little to the manuscript, and I suggest to remove from the text.

4. For the experiments in 3.5 CNN, you used respectively 90%, 5%, and 5% of the dataset for training, validation, and testing. Is it reasonable? May respectively 70%, 20%, and 10% for training, validation, and testing be more reasonable.

5. In the Results part, the author introduced one 10-layer CNN Architecture and tested three loss functions, i.e., MSE, MAE, and BCE, where the BCE achieved the best performance. Here, I would like to see the comparison with other state-of-the-art CNN models, to see the usability of the proposed method. If not possible, you can also discuss on that.

Author Response

  1. I suggest to give a framework of the methodology.
    We introduced a paragraph to give a framework and we introduced a chart with the correspondent description to clarify the workflow.

    Figure and paragraph were introduced in line 163 in the Methods section.
  2. I suggest to move “2.5. Convolutional Neural Network” to the “Method” Section.

    The 2.5. Convolutional Neural Network” in the Materials section only describe the used software and hardware while, in the Method section, the description of the architecture and hyperparameter was carried out. That is the reason why the authors believed that it would be clearer to separate them as it was done.

  3. In Subsection 3.2.2 for “post-processing”. You introduced two filter methods, i.e., the K-means method and the Mexican Hat wavelet method, where the latter is selected according to previous publication. That means, the K-means method contributes little to the manuscript, and I suggest to remove from the text.

    The two approaches are carried out sequentially, first the k-mean outlier removal and second the wavelet-based approach. Nevertheless, authors decided to remove Figure 3 since it doesn’t contribute much to the understanding of the method.

    A short sentence was added at line 219

    “Next, for the spike identification and outliers removal two approaches were combined sequentially.”

  4. For the experiments in 3.5 CNN, you used respectively 90%, 5%, and 5% of the dataset for training, validation, and testing. Is it reasonable? May respectively 70%, 20%, and 10% for training, validation, and testing be more reasonable.

    There is no general rule for the data splitting, it is true that usual ratios are close to the 70/20/10 or 80/10/10 rule. These ratios are usually accepted since the imposed task is typically require a high level of generalization, for instance in an ideal human faces detector, the dataset would require to include the entire population of humans in the world. In our case since the entire population is contained within the sample, we decided to use 90/5/5 splitting ratio checking that the validation and test set followed the training distribution. Therefore, we can assume that the three sets are equally representative, and the biggest amount of data is being used to train the model. The ratio influences the variance. With less testing data, the statistical performance of the model has a greater variance and if less training data is used the variance on the estimated parameters increases. Since the test set has a low standard deviation equal to 0.0422 we traded-off the statistical performance of the model for more robust weights of the parameters.

    The authors decided to include a comment in the line 332.

    “There is no general rule for the data splitting ratio. The splitting percentage has an influence on the variance. With less testing data, the statistical performance of the model has a greater variance and if less training data is used the variance on the estimated parameters increases. The choice of the splitting ratio was validated by checking the distribution similarity of the three sets. Therefore, the three sets can be assumed equally representative of the entire population, and the biggest amount of data is being used to train the model. Each set contains the corresponding tensors pairs of input (features) and ground truth (measured). The test set is used by the algorithm to learn by minimising the defined cost function, while the validation set is used to determine the optimal architecture and tune the hyper-parameters of the network. On the other hand, the test set is separated and never seen by the model. At evaluation time, we compare the estimated reconstruction with the true measured profile obtained with the profilometer.”

  5. In the Results part, the author introduced one 10-layer CNN Architecture and tested three loss functions, i.e., MSE, MAE, and BCE, where the BCE achieved the best performance. Here, I would like to see the comparison with other state-of-the-art CNN models, to see the usability of the proposed method. If not possible, you can also discuss on that.

    It was not possible to confront directly with other deep learning studies related to road surfaces since code and weights of the models are not public available at least to our knowledge. Regarding other deep learning architectures or types, CNNs are proven to be the best type of network type to deal with images. It was tested when the network was being developed against traditional MPL and the results were clearly outperforming by the CNNs. We also confronted against a typical encoder-decoder architecture known as U-net used for image translation and even if the results were similar, the size and number of parameters made the training process to slow without gaining any performance. The authors believed that including these results didn’t contribute to the manuscript and only showing the best architecture was more effective.

    Line 487 was considered in the section 5.3 Results.

    “It was not possible to compare the obtained results with other studies working with photometric deep learning for road surfaces, due to the unavailability of model weights to build and run the models. Another constrain to compare different methods lies in the fact that most of the current DL models require different input information or the expected ground truth are the normal directions. In the latter case, the ground truth used in this methodology correspond to laser profiles and the orientation of the faces are not provided, therefore the normal only have one component compared to the three expected for the other methods.”

Reviewer 2 Report

1. How to embody the main innovation of the article?

2. Are they compared with existing equipment and deep learning methods?

3. Can the measured road effect be supplemented?

4. It is suggested that the existing research methods should be investigated more fully.

Author Response

  1. How to embody the main innovation of the article?
    The main innovation of the article comes from the high accuracy achieved with the method, and most importantly, that no expertise in the current field (photometric stereo) is needed to develop the model. A new paragraph was added in line 104 to include the suggested comment.

    “Besides its scientific novelty, this paper brings an alternative solution to assess road surfaces, with two major contributions. First, no expertise is needed in the field of photometric, computer vision, or numerical optimisation to obtain the surface reconstruction. Second, the economical benefit obtained by using a commercial camera instead of a professional instrumentation.”
  1. Are they compared with existing equipment and deep learning methods?

    Yes, the results are compared with the “True” measured profile with the laser, the equipment has an accuracy of 0.05 [mm] on the vertical height as stated in Table 2, showing the high accuracy achieved by the method since is close to the one obtained with the laser.

    For the second point, it is not possible to confront directly with other deep learning studies related to road surfaces since code and weights of the models are not available to public at least to our knowledge. Regarding other deep learning architectures or types, CNNs are proven to be the best type of network type to deal with images. It was tested when the network was being developed against traditional MPL and the results were clearly outperformed by the CNNs. We also confronted against a typical encoder-decoder architecture known as U-net used for image translation and even if the results were similar, the size and number of parameters made the training process slower without gaining any performance. The authors believed that including these results didn’t contribute to the manuscript and only showing the best architecture was more effective.

    Nevertheless, line 487 was considered in the section 5.3 Results.

    “It was not possible to compare the obtained results with other studies working with photometric deep learning for road surfaces, due to the unavailability of model weights to build and run the models. Another constrain to compare different methods lies in the fact that most of the current DL models require different input information or the expected ground truth are the normal directions. In the latter case, the ground truth used in this methodology correspond to laser profiles and the orientation of the faces are not provided, therefore the normal only have one component compared to the three expected for the other methods.”

  1. Can the measured road effect be supplemented?
    I am sorry, I don’t understand the question. We are not measuring the road effect; we are measuring the road surface height. If the reviewer intended supplementing the new system with extra sensors or information. The answer would be, yes, we could, but the goal of the new developed system is to be low-cost, hence the fewest number of sensors are desired.
  1. It is suggested that the existing research methods should be investigated more fully.

    It is within our scope to explore the existing research methods and use them to compare our results. Nevertheless, as stated in the introduction the advantage of this methodology is that no expertise is needed in the specific field and anyone with basic deep learning knowledge can replicate the methodology on their own dataset an as shown in the paper achieve accurate results directly comparable with the laser profiler. Section Results was modified, and Section Discussion was created at line 421.

Reviewer 3 Report

The authors aim to develop the methodology to develop a fast and low-cost system using images taken with a commercial camera to recover the height information of the road surface using Convolutional Neural Networks. 

Overall, the paper is well developed and needs only minor improvements.

1. Authors should better elaborate on why such systems should be affordable to the broader public. E.g. they state, "Unfortunately, their high cost makes them unaffordable for a broader public." The validity of using such systems among the broader public should be discussed in the introduction. 

2. Since the main motivation is that the system should be available to the broader public, there should be a chapter before the conclusion in which the author should elaborate on the practical implications of their work. The construction of the device and the system for the broader public is the topic of a new scientific paper, but in this paper, at least the outline of such a system should be provided. 

3. The potential pitfalls of using an ad-hoc dataset should be addressed in the Materials section. 

4. Validation of the neural networks should be better elaborated. To be more precise, authors should provide a small discussion on which validity measures are used in similar systems, and why the chosen one are the best for the specific application. 

5. In the last section, please focus on “Discussion, Implication, and Conclusion” to include

(1).     Summary of the research - what was the goal, and how it was attained (2).     Discussion of why the authors found these results and how they comply (or not) with the Literature Review. (3).     Managerial Implications (4).     Limitations of the paper (5).     Future Studies and Recommendations

Author Response

  1. Authors should better elaborate on why such systems should be affordable to the broader public. E.g. they state, "Unfortunately, their high cost makes them unaffordable for a broader public." The validity of using such systems among the broader public should be discussed in the introduction.
    The second paragraph in the introduction was reformulated based on the suggested comment at line 23.

    “The assessment of road surfaces is a matter of concern to both the public administration and the private sector, either for research interests or for monitoring purposes. In order to carry out the task, several techniques have been studied over the years. Nowadays, contactless techniques are preferred over classical methods due to their portability and high precision. The high cost associated with the instrumentation makes them unaffordable for a broader public. Therein lies the importance to develop a simple, portable low-cost system that work with acceptable precision as an alternative solution to be adopted by these less favoured sectors. In this way, even if the economic resources are limited, the assessment of the road surface can still be made.”
  1. Since the main motivation is that the system should be available to the broader public, there should be a chapter before the conclusion in which the author should elaborate on the practical implications of their work. The construction of the device and the system for the broader public is the topic of a new scientific paper, but in this paper, at least the outline of such a system should be provided. 
    Even if the main motivation for the development of the system is as suggested by the reviewer, the scope of the paper was rather to proof the validity of the proposed methodology working with deep learning. Regarding the construction of the device, in the Methodology section, the outline for the system was explained. Nevertheless, the Section Discussions was created where the authors deepened as suggested by the reviewer the practical implications for the developed methodology.

    Section 5 was created at line 421 and the first paragraph in the Conclusion section was modified in Line 503

    “In the present paper, we started from the premise to develop a fast and low-cost system as an alternative solution to assess road surfaces for the least favoured sectors. The purpose was to obtain a faithful reconstruction of the 3D road surface. In order to do so, information coming from a sensor was needed. Keeping in mind the fast and low-cost objective, a commercial photographic camera was chosen due to its associated practical characteristics, i.e. compactness, simplicity to use, easy GUI, connection and remote control.”

    Also line 531 in the Conclusions section

    “The outcome showed that the results obtained with the proposed method are practically identical when these indicators are used. This entail an important practical application. The estimated indicators are practically the same as the ones calculated with the professional instrument, showing that the system yields excellent results for global assessment where global metrics are required.”

    Also line 543 in the Conclusions section

    “Besides the scientific novelty for the proposed method using CNN's, this paper brings an alternative solution to assess road surfaces, with three major contributions. First, no expertise is required in the field of photometric, computer vision, or numerical optimisation to obtain the surface reconstruction. Second, the associated economical reduction obtained using a commercial camera instead of a professional instrumentation. Third, the high accuracy achieved is remarkable.
    However the vast amount of data needed for the creation of the dataset as well as the associated time to train the network can be a significant. This encourages to pursuit a better generalised model working for all type of surfaces simultaneously. Nevertheless, the proposed methodology would stand, setting the base for this future work. This methodology could establish a new approach for texture measurement allowing a more affordable system for the least favoured sectors.”

  1. The potential pitfalls of using an ad-hoc dataset should be addressed in the Materials section.
    The drawback is not so much on the ad-hoc dataset rather in the methodology itself which works for one sample at the time, and this is stated at the Conclusions. We include this discussion in the new created section Discussions where the authors deepen as suggested by the reviewer the pitfalls of the ad-hoc dataset, the implications for the developed methodology and the results.
    Section 5 was created at line 420. Particularly the pitfalls of the dataset are highlighted in the subsection 5.1 Materials
  1. Validation of the neural networks should be better elaborated. To be more precise, authors should provide a small discussion on which validity measures are used in similar systems, and why the chosen one are the best for the specific application.
    Authors reformulated some paragraphs in the Results section, and created the Discussion section to deepen as suggested the typical indicators used in similar systems. Section 5 was created at line 420
  1. In the last section, please focus on “Discussion, Implication, and Conclusion” to include:
    (1).     Summary of the research - what was the goal, and how it was attained
    (2).     Discussion of why the authors found these results and how they comply (or not) with the Literature Review.
    (3).     Managerial Implications
    (4).     Limitations of the paper
    (5).     Future Studies and Recommendations

    The Section Discussion was created to fill the gaps that the reviewer exposed, also the Conclusion section was modified in order to include the remaining specific points asked by the reviewer.

Round 2

Reviewer 2 Report

It is suggested to supplement the references in recent years.